# A General Point-Based Method for Self-Calibration of Terrestrial Laser Scanners Considering Stochastic Information

**Tengfei Zhou [1]** , **Xiaojun Cheng [1,\*]**, **Peng Lin [2]**, **Zhenlun Wu [3]** and **Ensheng Liu [1,4]**

[1] College of Survey and Geo-Informatics, Tongji University, Shanghai 200092, China; 1710639@tongji.edu.cn (T.Z.); 1410893@tongji.edu.cn (E.L.)

[2] College of Civil Engineering, Anhui Jianzhu University, Hefei 232001, China; penglin1991@ahjzu.edu.cn

[3] Big Data Development Administration of Yichun, Yichun 336000, China; ycwzl@yichun.gov.cn

[4] College of Building Engineering, Jing Gang Shan University, Ji'an 343009, China

\* Correspondence: cxj@tongji.edu.cn

**Abstract:** Due to the existence of environmental or human factors, and because of the instrument itself, there are many uncertainties in point clouds, which directly affect the data quality and the accuracy of subsequent processing, such as point cloud segmentation, 3D modeling, etc. In this paper, to address this problem, stochastic information of point cloud coordinates is taken into account, and on the basis of the scanner observation principle within the Gauss–Helmert model, a novel general point-based self-calibration method is developed for terrestrial laser scanners, incorporating both five additional parameters and six exterior orientation parameters. For cases where the instrument accuracy is different from the nominal ones, the variance component estimation algorithm is implemented for reweighting the outliers after the residual errors of observations obtained. Considering that the proposed method essentially is a nonlinear model, the Gauss–Newton iteration method is applied to derive the solutions of additional parameters and exterior orientation parameters. We conducted experiments using simulated and real data and compared them with those two existing methods. The experimental results showed that the proposed method could improve the point accuracy from $10^{-4}$ to $10^{-8}$ (a priori known) and $10^{-7}$ (a priori unknown), and reduced the correlation among the parameters (approximately 60% of volume). However, it is undeniable that some correlations increased instead, which is the limitation of the general method.

**Keywords:** self-calibration; Gauss–Helmert model; random error; Gauss–Newton method; variance component estimation

## 1. Introduction

In contrast to the traditional single-point acquisition method, terrestrial laser scanning (TLS) technology greatly improves work efficiency with a variety of applications [1–4]. While during the procedure of point cloud data acquisition, TLS could irresistibly be affected by, e.g., the instrument itself, the external environment, the scanning targets, etc., which results in the point cloud coordinates being modified by the systematic and random errors to varying degrees, reducing the observation accuracy of point cloud coordinates to a certain extent. Consequently, the coordinates obtained by TLS and the real coordinates of the target points are not always corresponded.

Similar to the total station (TS), three-dimensional coordinates in laser point clouds are calculated via the spherical coordinate system subjected to the oblique distances, horizontal, and vertical angles measured by the instruments itself. On the other side, systematic errors, in the course of scanning, caused by ranging the angle of incident, target reflectivity, and temperature are undoubtedly not

negligible [5]. All these above factors directly affect the accuracy of point cloud data [6], and to some extent, weaken the accuracy of subsequent point cloud processing. Fortunately, users can normally correct or evaluate the results of the scanning measurement, according to the application environment and nominal accuracy of the instrument from the manufacturer. However, in the event that the nominal accuracy of the instrument loses consistency with the actual ones, the above operations may yield incorrect results, which is, in fact, extremely usual. It is, therefore, crucial to make a reasonable determination of the actual accuracy (or additional parameters), the so-called calibration (or self-calibration) of the instruments correctly and rationally.

The traditional calibration methods [7–10], which require more demanding conditions on the environment and operators, are to observe the determined targets in the angles and distances, separately to acquire the additional parameters (APs) containing the addition constant, horizontal axis error, etc. In recent years, self-calibration methods, bringing unknown parameters into the function model for calculation, can be divided into two categories—point-based and plane-based—have attracted the attention of researchers in TLS calibrations. In the case of point-based methods, it is typically a matter of fitting hundreds (thousands) of observations to the center of the target object to obtain point features [11].

Gielsdorf et al. [12] pioneered the concept of TLS calibration for low-cost scanners. Lichti [13–16] presented a rigorous method for self-calibration of TLS using a network of signalized points by adding a set of APs to the spherical coordinate observation equation, enabling the calibration of the amplitude-modulated-continuous-wave (AM-CW) scanner system [13]. Schneider [17] used the approach proposed in [13] to fulfill the self-calibration of Riegl LMS-Z420i (Riegl, 2007). Reshetyuk [18] implemented the calibration of Callidus 1.1 (Callidus, 2002), HDS (High Definition Surveying) 3000 (Leica, 2003), and HDS 2500 (Leica, 2001) in a specially designed indoor 3D calibration facility. Afterwards Reshetyuk et al. [19] constructed a function model by associating appropriate weights to the observations with the point clouds directly georeferenced, adopting the notation from [13], to weaken the correlations among the parameters [20,21]. Lichti [22] presented the full mathematical model for a point-based photogrammetric approach to implement the FARO LS880 (FARO, 2007) self-calibration. Lerma [23] proposed a method for determining the optimal set of additional parameters to achieve a priori unknown systematic errors modeling based on a dimensionless quality index. Medic [24] studied an empirical stochastic model based on point feature self-calibration, and addressed the problem of the stochastic model as well as the factual incompatibility by examining the uncertainty of the scanner target point.

However, point-based self-calibration methods often make the parameters correlated with each other, while one goal of the self-calibration network design is to reduce the functional dependence in model variables. Thereby, the point-based self-calibration methods often require the deployment of signalized points covering the entire field of view [10,19]. Admittedly, the point-based methods are cumbersome to operate and demanding for the observing environment. On the other hand, plane-based self-calibration methods are more adaptable and have been implemented by [12,25–29].

Looking at the recent self-calibration literature, it can be found that some existing methods [30–32] are very similar in terms of function models, i.e., Equation (4). They do not take into account random errors, which to some extent cause the instability of the function models and make it impossible to avoid the influence of random errors on the models itself and on the parameter solutions. In other words, due to the fact that the random errors of the observed values are not included in the function model, this can lead to the possibility of mixing a part of the random errors in the estimated APs, so that the parameter estimates do not correspond to the actual situation, and thus, the accuracy of the parameter solution cannot be guaranteed. In addition, there are still multiple scientific publications that consider the nominal accuracy as a criterion for weighting, or impose some kinds of constraints to the parameters [13,19,25]. As a result, the existing calibration models are not theoretically rigorous or inadvertently increase the complexity of the solution. The nonlinear Gauss–Helmert (GH) model [33], verified by [34,35], has no restriction on the form of functional relationship among the quantities

involved in the model [36,37], which is an effective solution to avoid the above problems. In reality, distances and angles are two types of observations with diverse units. Moreover, the difference between the nominal and real accuracy of the instrument makes it unreasonable to use equal or direct weights. It's a proper time to introduce the variance component estimation (VCE) algorithm [38–40], which provides a solution for this plight.

Based on the observation equation of the TLS and GH models (spatial transformation model for specific), a general self-calibration model of the scanner with 11 parameters is constructed, including three translation parameters and three rotation parameters (exterior orientation parameters, EOPs), together with five instrument system error parameters (five APs). Due to the nonlinear nature of the general method, the Gauss–Newton iteration algorithm in [41] is employed to derive the solution of APs and EOPs. Furthermore, in the procedure of data processing, both optimal parameter estimation and reasonable accuracy assessment, are conducted based on the stochastic model (variance information) observations as the premise. As a result, ignoring random information or using a priori accuracy for weighting does not yield a reasonable stochastic model, and will generate an adverse impact on the parameter estimation. Finally, the VCE algorithm is applied to correct the weights after obtaining the residuals of the observations. Overall, compared to other studies, the general self-calibration method can account for random errors in the observations so the function model is more rigorous, and updating the covariance matrix using the VCE theory yields more accurate estimates of the parameters.

The rest of the paper is organized as follows. In Section 2, the observation principle of the scanner and the general self-calibration model are first introduced, and then the derivation procedure is described in detail. The experiments are presented in Section 3. We present results and discussions in Section 4. Finally, we conclude the paper with a summary of our work in Section 5.

## 2. Methods

### 2.1. Observation Principle of TLS

TLS establishes its datum on the basis of an independent left-handed coordinate system with the original point **O** located in the center of the scanner. The X-axis is in the transverse scanning plane, while the Y-axis is perpendicular to the X-axis, and the Z-axis is perpendicular to the X-Y plane. The original observation data of the TLS [12,20] in the spherical coordinate system are the oblique distance $s$, vertical angle $\theta$, and horizontal angle $\alpha$, i.e., $(s, \theta, \alpha)$, as shown in Figure 1.

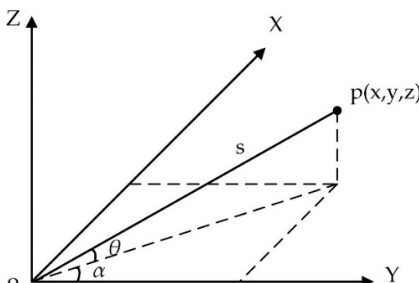

**Figure 1.** Observation principle of TLS.

For the purpose of obtaining the three-dimensional coordinates of the target point relative to the origin, it is necessary to transform the original observations into the Cartesian coordinate system. As illustrated in Equation (1), the transformation method for the above two coordinate systems, i.e., the observation principle of TLS [13,16,20] is shown.

$$\begin{cases} x = s \cdot \cos\theta \cdot \cos\alpha \\ y = s \cdot \mathrm{ccos}\theta \cdot \sin\alpha \\ z = s \cdot \sin\theta \end{cases} \tag{1}$$

where $[x, y, z]^{\mathrm{T}}$ denotes the Cartesian coordinate vector of a single point obtained by TLS.

Likewise, the distance and angle data required for the self-calibration could also be converted out from Cartesian coordinates through Equation (2) [28].

$$\begin{cases} s = \sqrt{x^2 + y^2 + z^2} \\ \alpha = \tan^{-1}(y/x) \\ \theta = \tan^{-1}\!\left(z / \sqrt{x^2 + y^2}\right) \end{cases} \tag{2}$$

From the observation equation, i.e., Equation (1), it is clear that the distances and angles are the direct observations. In addition to random errors, their systematic errors can also affect the scanning results, even if the target and environmental factors are typically different. Since the scanners are based on the optical ranging principle of distance measurement, with reference to the TS, there are two main types of APs in them, that is, the addition constant $m$ and the multiplication constant $\lambda$, caused by mistakes in instrument manufacturing and installation, in conjunction with operational variations. As for the three APs of the angles, collimation error $c$ and horizontal axis error $i$ are often present in horizontal angle observations, toward the vertical index error $t$ [16,18,28]. The correction results of the collimation axis error $c'$ and horizontal axis error $i'$ to the horizontal angle observations follow the equation:

$$\begin{cases} c' = c / \cos\theta \\ i' = i \cdot \tan\theta \end{cases} \tag{3}$$

## 2.2. General Self-Calibration Model

The above-mentioned reasons cause the obtained coordinate observations to not correspond to the real values, so a reasonable calibration method is necessary to effectively remove APs. Conventional self-calibration methods apply high-precision instruments (e.g., TS) and scanners to measure homonymous points uniformly distributed in space (e.g., target or target sphere) to collect the coordinates of the respective datum, after which the TS's measurement points are treated as a reference. Possible existing APs are considered as parameters to be estimated and brought into the model for adjusting [19,22,30]. Accordingly, the self-calibration function model is usually represented as a variant of the non-line Gauss–Markov (GM) model [42]. Processing the coordinate sequence through the GM model is to solve the vector of unknown parameters, containing five APs and six EOPs, which results in the identities;

$$\begin{bmatrix} X \\ Y \\ Z \end{bmatrix} = \mathbf{R}\begin{bmatrix} x \\ y \\ z \end{bmatrix} + \begin{bmatrix} \Delta x \\ \Delta y \\ \Delta z \end{bmatrix} = \mathbf{R}\begin{bmatrix} [s \cdot (1+\lambda) + m] \cdot \cos(\theta + t) \cdot \cos(\alpha + c' + i') \\ [s \cdot (1+\lambda) + m] \cdot \cos(\theta + t) \cdot \sin(\alpha + c' + i') \\ [s \cdot (1+\lambda) + m] \cdot \sin(\theta + t) \end{bmatrix} + \begin{bmatrix} \Delta x \\ \Delta y \\ \Delta z \end{bmatrix} \tag{4}$$

where $[X, Y, Z]^{\mathrm{T}}$ denotes the coordinate vector of homonymous points observed through TS. $[\Delta x, \Delta y, \Delta z]^{\mathrm{T}}$ represent the translation parameters. $[m, \lambda, c', i', t]^{\mathrm{T}}$ represent the APs. $\mathbf{R}$ is the rotation matrix, including three parameters, in the form of:

$$\mathbf{R} = \mathbf{R}_\varphi \mathbf{R}_\omega \mathbf{R}_\kappa = \begin{bmatrix} \cos\varphi & 0 & -\sin\varphi \\ 0 & 1 & 0 \\ \sin\varphi & 0 & \cos\varphi \end{bmatrix}\begin{bmatrix} 1 & 0 & 0 \\ 0 & \cos\omega & -\sin\omega \\ 0 & \sin\omega & \cos\omega \end{bmatrix}\begin{bmatrix} \cos\kappa & -\sin\kappa & 0 \\ \sin\kappa & \cos\kappa & 0 \\ 0 & 0 & 1 \end{bmatrix} \tag{5}$$

where $(\varphi, \omega, \kappa)$ are the parameters geared to the rotation vector, i.e., the Euler angles rotating around the Y-axis, the X-axis, and the Z-axis, respectively. Regarding Equation (4), $(\Delta x, \Delta y, \Delta z, \varphi, \omega, \kappa)$ are the EOPs of the general self-calibration method.

For nonlinear models, Equation (4) generally needs to be linearized before solving for the unknown parameters [41]. Considering that the principle of indirect adjustment, the following error equation can be written

$$V = A\xi - L \tag{6}$$

Here, $V$ and $A$ denote the residual error vector and coefficient matrix of the parameter vector, respectively. $\xi$ is the parameter vector to be estimated, and $L$ indicates the observation vector.

In the absence of weights, the objective function to be minimized is obtained in the form:

$$V^T V = \min \tag{7}$$

Starting from this state of discussion, and taking random errors into consideration, the present study aims to achieve the following objectives:

1. Weight observations according to its corresponding prior information to solve the unknown parameters, for the sake of attenuating the effect of random errors on the coordinates of TLS;
2. For cases where the actual accuracy differs from the nominal accuracy, a posteriori determination of the observed values is performed on the basis of the VCE algorithm.
3. Develop a general self-calibration model for the scanner and derive its solution based on the nonlinear GH model and instrumental measurement principle within the weighted total least square algorithm (WTLS);

The general point-based method for the TLS self-calibration proposed in this paper is constructed in the form of the GH model. Its specific representation is similar to Equation (4), but takes the random error of the original observations into account within the function model.

$$\begin{bmatrix} X \\ Y \\ Z \end{bmatrix} = R \begin{bmatrix} [(s - e_s) \cdot (1 + \lambda) + m] \cdot \cos(\theta - e_\theta + t) \cdot \cos(\alpha - e_\alpha + c' + i') \\ [(s - e_s) \cdot (1 + \lambda) + m] \cdot \cos(\theta - e_\theta + t) \cdot \sin(\alpha - e_\alpha + c' + i') \\ [(s - e_s) \cdot (1 + \lambda) + m] \cdot \sin(\theta - e_\theta + t) \end{bmatrix} + \begin{bmatrix} \Delta x \\ \Delta y \\ \Delta z \end{bmatrix} \tag{8}$$

To one single point, where $(e_s, e_\theta, e_\alpha)$ denotes the random error vector of the distance and angles in the vertical and horizontal directions; the meanings of those remaining can be referred to Equations (4) and (5).

The stochastic model for the general self-calibration method can be written in the form;

$$e = \begin{bmatrix} e_s \\ e_\theta \\ e_\alpha \end{bmatrix} \sim \left( \begin{bmatrix} 0 \\ 0 \\ 0 \end{bmatrix}, \begin{bmatrix} \sigma_s^2 & 0 & 0 \\ 0 & \sigma_\theta^2 & 0 \\ 0 & 0 & \sigma_\alpha^2 \end{bmatrix} \right) \tag{9}$$

with the objective function:

$$e^T P e = \min \tag{10}$$

where $(\sigma_s, \sigma_\theta, \sigma_\alpha)$ represent median errors in the distance and angles, respectively.

The corresponding parameter vector can be defined as

$$\xi = \begin{bmatrix} \Delta x & \Delta y & \Delta z & \varphi & \omega & \kappa & m & \lambda & c & i & t \end{bmatrix}^T \tag{11}$$

### 2.3. Derivation of General Self-Calibration Model

Since the proposed self-calibration model is essentially a nonlinear model, the corresponding estimates can no longer be claimed as least squares (LS) estimates. Due to the nonlinear nature of the Equation (8), the Gauss–Newton method [41] of the nonlinear LS is adopted to derive the

solution. We assume that the appropriate approximate values (initial values) of $e$ are $e^0 = (e_s^0, e_\theta^0, e_\alpha^0)$. The unknown parameter vector approximate values are set up as:

$$\xi^0 = \begin{bmatrix} \Delta x^0 & \Delta y^0 & \Delta z^0 & \varphi^0 & \omega^0 & \kappa^0 & m^0 & \lambda^0 & c^0 & i^0 & t^0 \end{bmatrix}^{\text{T}} \tag{12}$$

For the initial values of the parameter vector, the linear GM model can be utilized to solve the EOPs. In addition, we can deem that the instrument is the ideal state at the time of the manufacturing or measurement so that the initial values of the APs and the residual vector of observations can be considered as *Zero* (a matrix with all elements zero) [19,30,35]. By substituting

$$H = \begin{bmatrix} x \\ y \\ z \end{bmatrix} = \begin{bmatrix} [(s-e_s) \cdot (1+\lambda) + m] \cdot \cos(\theta - e_\theta + t) \cdot \cos(\alpha - e_\alpha + c' + i') \\ [(s-e_s) \cdot (1+\lambda) + m] \cdot \cos(\theta - e_\theta + t) \cdot \sin(\alpha - e_\alpha + c' + i') \\ [(s-e_s) \cdot (1+\lambda) + m] \cdot \sin(\theta - e_\theta + t) \end{bmatrix} \tag{13}$$

The right-hand members of Equation (8) are expanded at $(\xi^0, e^0)$ through the binary Taylor series, and the yield of the linear equation expressed within new parameters:

$$\begin{bmatrix} X \\ Y \\ Z \end{bmatrix} = R^j H^j + \begin{bmatrix} \Delta x^j \\ \Delta y^j \\ \Delta z^j \end{bmatrix} + \begin{bmatrix} d\Delta x \\ d\Delta y \\ d\Delta z \end{bmatrix} + \frac{\partial R^j}{\partial \varphi} H^j d\varphi + \frac{\partial R^j}{\partial \omega} H^j d\omega + \frac{\partial R^j}{\partial \kappa} H^j d\kappa + R^j \frac{\partial H^j}{\partial m} dm +$$
$$R^j \frac{\partial H^j}{\partial \lambda} d\lambda + R^j \frac{\partial H^j}{\partial c} cc + R^j \frac{\partial H^j}{\partial i} di + R^j \frac{\partial H^j}{\partial t} dt + R^j \frac{\partial H^j}{\partial e} \left( e - e^j \right) \tag{14}$$

where $j$ the superscript denotes the sequence number of the iteration; during the first iteration, the matrices involved can be populated with the initial values; $\partial$ is the symbol of the partial derivative, where:

$$\frac{\partial R^j}{\partial \varphi} = \frac{\partial R_\varphi^j}{\partial \varphi} R_\omega^j R_\kappa^j = \begin{bmatrix} -R^j(3,1) & -R^j(3,2) & -R^j(3,3) \\ 0 & 0 & 0 \\ R^j(1,1) & R^j(1,2) & R^j(1,3) \end{bmatrix} \tag{15}$$

$$\frac{\partial R^j}{\partial \omega} = R_\varphi^j \frac{\partial R_\omega^j}{\partial \omega} R_\kappa^j = \begin{bmatrix} -\sin\varphi^j R^j(2,1) & -\sin\varphi^j R^j(2,2) & -\sin\varphi^j R^j(2,3) \\ -\sin\omega^j \cos\kappa^j & -\sin\omega^j \sin\kappa^j & -\cos\omega^j \\ \cos\varphi^j R^j(2,1) & \cos\varphi^j R^j(2,2) & \cos\varphi^j R^j(2,3) \end{bmatrix} \tag{16}$$

$$\frac{\partial R^j}{\partial \kappa} = R_\varphi^j R_\omega^j \frac{\partial R_\kappa^j}{\partial \kappa} = \begin{bmatrix} -R^j(1,2) & R^j(1,1) & 0 \\ -R^j(2,2) & R^j(2,1) & 0 \\ -R^j(3,2) & R^j(3,1) & 0 \end{bmatrix} \tag{17}$$

Substituting

$$\begin{cases} s^j = (s - e_s^j) \cdot (1 + \lambda^j) + m^j \\ \theta^j = \theta - e_\theta^j + t^j \\ \alpha^j = \alpha - e_\alpha^j + c'^j + i'^j \end{cases} \tag{18}$$

The coefficient matrix of the APs vector can be expressed as:

$$\begin{cases} \frac{\partial H^j}{\partial m} = \begin{bmatrix} \cos\theta^j \cdot \cos\alpha^j \\ \cos\theta^j \cdot \sin\alpha^j \\ \sin\theta^j \end{bmatrix}, \frac{\partial H^j}{\partial \lambda} = \begin{bmatrix} \cos\theta^j \cdot \cos\alpha^j \\ \cos\theta^j \cdot \sin\alpha^j \\ \sin\theta^j \end{bmatrix} \cdot (s - e_s^j), \frac{\partial H^j}{\partial t} = \begin{bmatrix} -s^j \cdot \sin\theta^j \cdot \cos\alpha^j \\ -s^j \cdot \sin\theta^j \cdot \sin\alpha^j \\ s^j \cdot \cos\theta^j \end{bmatrix} \\ \frac{\partial H^j}{\partial c} = \begin{bmatrix} -s^j \cdot \cos\theta^j \cdot \sin\alpha^j \\ s^j \cdot \cos\theta^j \cdot \cos\alpha^j \\ 0 \end{bmatrix} \cdot \frac{1}{\cos\theta^j}, \frac{\partial H^j}{\partial i} = \begin{bmatrix} -s^j \cdot \cos\theta^j \cdot \sin\alpha^j \\ s^j \cdot \cos\theta^j \cdot \cos\alpha^j \\ 0 \end{bmatrix} \cdot \tan\theta^j \end{cases} \tag{19}$$

As with the parameters solution, the observation equation considering the random errors, namely Equation (13), is a nonlinear model that also requires a partial derivative for each random error residual

of the distance and angles in the form of $\left[e_s - e_s^0, e_\theta - e_\theta^0, e_\alpha - e_\alpha^0\right]^{\mathrm{T}}$; thus the corresponding residual coefficient matrix can be represented as:

$$\frac{\partial H^j}{\partial e} = \begin{bmatrix} -(1+\lambda^j)\cdot\cos\theta^j\cdot\cos\alpha^j & s^j\cdot(\cos\theta^j\cdot\sin\alpha^j\cdot\gamma+\sin\theta^j\cdot\cos\alpha^j) & s^j\cdot\cos\theta^j\cdot\sin\alpha^j \\ -(1+\lambda^j)\cdot\cos\theta^j\cdot\sin\alpha^j & s^j\cdot(-\cos\theta^j\cdot\cos\alpha^j\cdot\gamma+\sin\theta^j\cdot\sin\alpha^j) & -s^j\cdot\cos\theta^j\cdot\cos\alpha^j \\ -(1+\lambda^j)\cdot\sin\theta^j & -s^j\cdot\cos\theta^j & 0 \end{bmatrix} \quad (20)$$

where

$$\gamma = c\cdot\sec\theta^j\cdot\tan\theta^j + i\cdot\sec^2\theta^j \quad (21)$$

The altered vector of the new parameters to be estimated is:

$$\mathrm{d}\xi = \begin{bmatrix} \mathrm{d}\Delta x & \mathrm{d}\Delta y & \mathrm{d}\Delta z & \mathrm{d}\varphi & \mathrm{d}\omega & \mathrm{d}\kappa & \mathrm{d}m & \mathrm{d}\lambda & \mathrm{d}c & \mathrm{d}i & \mathrm{d}t \end{bmatrix}^{\mathrm{T}} \quad (22)$$

For ease of comprehension and reading, it is capable to merge similar items of the function model, resulting in

$$X = R^j H^j + \Delta X^j + A^j \mathrm{d}\xi + B^j(e - e^j) \quad (23)$$

Here, $A^j$ represents the coefficient matrix of the parameters, with the form

$$A^j = \begin{bmatrix} E_{3\times3} & \frac{\partial R^j}{\partial \varphi}H^j & \frac{\partial R^j}{\partial \omega}H^j & \frac{\partial R^j}{\partial \kappa}H^j & R^j\frac{\partial H^j}{\partial m} & R^j\frac{\partial H^j}{\partial \lambda} & R^j\frac{\partial H^j}{\partial c} & R^j\frac{\partial H^j}{\partial i} & R^j\frac{\partial H^j}{\partial t} \end{bmatrix} \quad (24)$$

$$X = \begin{bmatrix} X \\ Y \\ Z \end{bmatrix}, \Delta X^j = \begin{bmatrix} \Delta x^j \\ \Delta y^j \\ \Delta z^j \end{bmatrix}, B^j = R^j\frac{\partial H^j}{\partial e} \quad (25)$$

where $E_{3\times3}$ is the unit matrix with three dimensions.

After linearization of the general self-calibration model, we draw on the Gauss–Newton method to implement the solving of unknown parameters and residuals of observations.

Substituting:

$$L^j = X - R^j H^j - \Delta X^j + B^j e^j \quad (26)$$

Establishing contact with the weight matrix $P$, the Lagrange objective function can be constructed in the identity:

$$\Phi = e^{\mathrm{T}}Pe + 2K^{\mathrm{T}}\left(L^j - A^j\mathrm{d}\xi - B^je\right) = \min \quad (27)$$

where $K$ is a vector of the auxiliary "Lagrange multipliers". The weight matrix $P$ can be represented as follows:

$$P = blkdiag\left(\frac{\sigma_0^2}{\sigma_{s_q}^2}, \frac{\sigma_0^2}{\sigma_{\theta_q}^2}, \frac{\sigma_0^2}{\sigma_{\alpha_q}^2}\right) \quad (28)$$

where $q$ ranges from 1 to $n$; $n$ denotes the total number of homonymous points; $(\sigma_s, \sigma_\theta, \sigma_\alpha)$ representing a priori information, which can be obtained from the nominal accuracy.

The solution of this objective function can be derived by means of the Euler–Lagrange necessary conditions, i.e., the partial derivatives of each variable are equal to zero. We can readily obtain the correction vector of the unknown parameters and residual error vector of the observations as follows:

$$\mathrm{d}\hat{\xi}^j = \left(\left(A^j\right)^{\mathrm{T}}\left(Q_c^j\right)^{-1}A^j\right)^{-1}\left(A^j\right)^{\mathrm{T}}\left(Q_c^j\right)^{-1}L^j \quad (29)$$

$$\hat{e}^j = Q\left(B^j\right)^{\mathrm{T}}K \quad (30)$$

where '^' indicates the estimation value; $Q$ is the cofactor matrix of the observations, and $Q = P^{-1}$;

$$Q_c^j = B^j Q \left(B^j\right)^{\mathrm{T}} \tag{31}$$

Thereby, the random error vector $\hat{e}^j$ of the observations needs to be updated in each iteration according to Equation (30), and the parameter vectors and random errors of the observation vectors after the first $(j+1)$ iteration are updated as:

$$\widetilde{\xi}^{j+1} = \widetilde{\xi}^j + \mathrm{d}\hat{\xi}^j \tag{32}$$

Here, '~' indicates the prediction value; after stripping the solution $\widetilde{\xi}^{j+1}$ and $\hat{e}^{j+1}$ of its random character, it is then used in the next iteration step as the approximation [34,35,41], which also shows that the initial values in Equation (12) can only be used in the first iteration. As a consequence, the mean square error of the unit weight, and the covariance matrix of the estimated parameters can be estimated via:

$$\hat{\sigma}_0 = \sqrt{e^{\mathrm{T}} P e / (3n - 11)} \tag{33}$$

$$D_x = \hat{\sigma}_0^2 \left(\left(A^j\right)^T \left(Q_c^j\right)^{-1} A^j\right)^{-1} \tag{34}$$

However, the nominal accuracy of an observed value is often not equal to the actual ones [41,42], i.e., the observed value at the first adjustment given the weights are essentially inappropriate, which would make the results obtained incorrect, although it may be very close to the true value. Therefore, a posterior estimation is capable of solving unknowns and correcting observations. As the raw observations in the general self-calibration model include both distances and angles, it is a reasonable time to introduce the VCE. Similar to the Gauss–Newton method, the VCE algorithm is iterative and terminates when the weight ratios of the various types of observations converge to one [40–42]. In this case, the corresponding residual vector and its coefficient matrix, including the corresponding weights and covariance matrix are altered, mainly a change in the position of the elements in the matrix; see Appendix A for specific.

The main purpose of the general self-calibration method is to restore the true positions of the object's surface coordinates in space by removing uncertainties in point clouds, i.e., random and systematic errors, and to provide a basis for improving subsequent point cloud segmentation [43,44], 3D modeling [45,46], etc. The general self-calibration method of TLS can be realized through the following steps:

1. Determining the iterative initial value of the unknown parameters.

   The linear transformation model would be appropriately adopted to obtain the initial values of EOPs, as described in the first identity of Equation (4), or take a simpler assumption that the translation parameter and rotation parameter are set to *Zero*, which will inevitably increase the number of iterations or the convergence time. It's important to note that doing so (EOPs $\propto$ *Zero*) risks converging to failure. In terms of APs, it can be considered that all kinds of errors are completely eliminated in the manufacture and installation of the instrument, so that the initial value of the system errors can be set to *Zero* [19,30,35].

2. Computing original observations of homonymous point as illustrated in Figure 1.

   The original observations, i.e., distances and angles, on the basis of the 3D coordinate data acquired by the scanner, are deduced according to Equation (2). Vice versa, the 3D coordinates can be back-calculated for comparing or checking, if needed.

3. Iterative process

(1)    populating the matrices $A^j, Q_c^j, L^j, B^j, P^j$, respectively, according to Equations (24)–(26), (28) and (31);

(2)    Predicting the residual error vector of the parameters and random errors in the first $j_{th}$ iteration through Equations (29) and (30);

(3)    With the help of the initial value, $\widetilde{\xi^j}$ can be updated via Equation (32) to obtain matrices mentioned in step (1) for the next iteration;

(4)    Steps (1)–(3) are to be repeated until the break-off conditions are achieved [33,35], and terminate the iteration;

(5)    Repeating steps (1)–(4) to reweight the observations based on $\hat{e}^j$, using VCE until convergence.

4.    Accuracy evaluation.

From Equations (33) and (34), the standard deviation to the homonymous points can be deduced, toward the covariance information.

It is important to note that the matrices involved in step (1) are updated imperatively during every iteration, otherwise there will be no convergence [35,41]. Also, the vector $L$ can be set at the initial value during the first iteration, but it is not valid for all the later iterations.

Since the general self-calibration method essentially is a nonlinear model, and the derivation process is relatively complex, to facilitate the reader's understanding, the overall comprehension as well as reproduction, a flowchart is additionally created for the description of the individual steps, as shown in Figure 2.

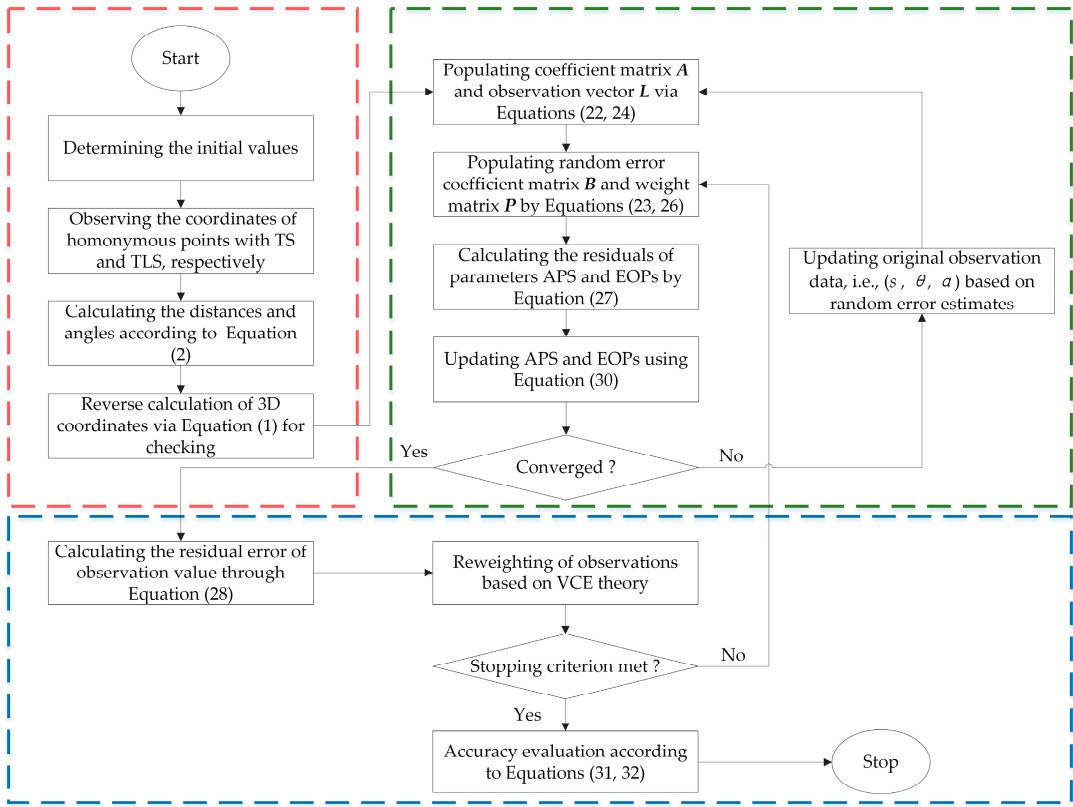

**Figure 2.** A flowchart of the general self-calibration method. The red part indicates the process of initialization and coordinate conversion, while the green and blue parts indicate the a priori processing and a posteriori estimation procedure of the proposed general self-calibration method, respectively.

## 3. Experiments

The experiments of the general method were carried out using simulated and real data. The data in the simulated experiment were derived from practical instrument parameters, with the external environmental influences such as atmospheric parameters, temperature and humidity neglected, as well as some parameters related to the instrumentation leveling, which can be found in [13,15,19]. In addition, there is often a correlation wandering in unknown parameters [19–21], which is along with the network design. The experimental idea was to use a high-precision TS and a scanner to be calibrated to obtain observation data of the points at the specified location, respectively. For the two sets of acquired coordinate data, we assumed the TS data as a reference, i.e., without containing any systematic nor random errors. All the experiments were performed in MatlabR2019b [47], which focused on the coordinate simulation and data processing. The proposed general self-calibration model was implemented to calculate the APs and EOPs compared with two existing methods.

### 3.1. Simulated Data

Assuming that there are 80 points distributed randomly in a space domain, 70 of which are homonymous, and the remaining 10 are used for checking. The distribution of these points is generated based on the real scanner's field of view (FOV), where the distances are set from 2 to 30 m; the horizontal angles are set from 0 to 360° and the vertical angles are varied from −45° to 90°. The corresponding standardized residual, following a zero-mean and unit variance Gaussian density function [25,44], to each true value need to be generated and added on the basis of the a priori standard deviations, conducted under the standard null hypothesis with 4 mm for distances and 0.0033° for horizontal and vertical angles, respectively. This simulation loops 5000 times.

Through Equation (1), we can convert the target points from a spherical coordinate system to a Cartesian coordinate system. Taking the results of one of the simulations as an example to illustrate the distribution of the target points in the TLS space, the results are shown in Figure 3.

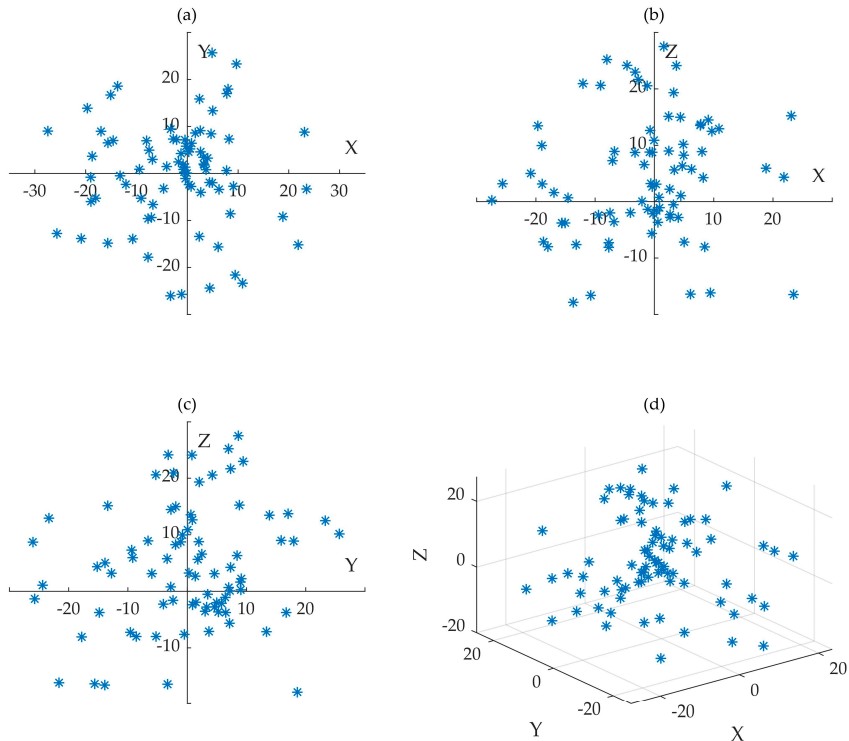

**Figure 3.** Distribution of point cloud Cartesian coordinates in the TLS system. (**a**) Projection to X-Y plane; (**b**) Projection to X-Z plane; (**c**) Projection to Y-Z plane; (**d**) Cartesian coordinate system.

As for the unknown parameters, we stipulated the true value of EOPs and APs, which are listed in Table 1, where the specific connotations of APs and EOPs can be found in the notes to Equations (4) and (5).

**Table 1.** True values of EOPs and APs in simulation experiment.

| EOPS | | | | | | | APs | | | |
|---|---|---|---|---|---|---|---|---|---|---|
| $\Delta x/\text{m}$ | $\Delta y/\text{m}$ | $\Delta z/\text{m}$ | $\varphi/\text{rad}$ | $\omega/\text{rad}$ | $\kappa/\text{rad}$ | $m/\text{m}$ | $\lambda$ | $c/\text{rad}$ | $i/\text{rad}$ | $t/\text{rad}$ |
| 5 | 10 | 5 | 0.2 | −0.2 | 1.0 | 0.005 | $10^{-4}$ | −0.01 | $10^{-3}$ | $-10^{-5}$ |

According to the scanner observation principle, i.e., Equation (1), the 3D coordinates of the scanner data could be calculated toward the TS subjecting to the first identity of Equation (4) by ignoring the APs.

The following three strategies were employed to implement the self-calibration of TLS.

1.  Nonlinear least-squares without regard to systematic errors;
2.  Self-calibration method based on nonlinear least-squares ignoring the random errors;
3.  General self-calibration method proposed in this paper.

At the beginning of the adjustment, it was assumed that the square root of the a priori variance component $\sigma_0$ was 0.001, and all the observation values in two datum were expected to be uncorrelated. We calculated the root mean square error (RMSE) for the $x$, $y$, and $z$ of 70 homonymous points, and the remaining 10 points by the following identities:

$$\sigma_x = \sqrt{\sum_{q=1}^{70} \left(\widetilde{x} - x_{\text{total}}\right)^2 / 70} \tag{35}$$

$$\sigma_y = \sqrt{\sum_{q=1}^{70} \left(\widetilde{y} - y_{\text{total}}\right)^2 / 70} \tag{36}$$

$$\sigma_z = \sqrt{\sum_{q=1}^{70} \left(\widetilde{z} - z_{\text{total}}\right)^2 / 70} \tag{37}$$

Therefore, the positional RMSE was derived from:

$$\sigma_p = \sqrt{\sigma_x^2 + \sigma_y^2 + \sigma_z^2} \tag{38}$$

where 'total' represents the observations from the TS; '~' here indicates the corrected coordinates calculated by Equation (13). RMSE for the checking points is similar to the above Equations (33)–(36), just replacing the denominator into the total number of remaining points.

Accordingly, we could calculate the differences between the adjustment values and the true values of the parameters to obtain the RMSE of each parameter as well

$$\nabla \xi = \widetilde{\xi} - \xi_{\text{true}} \tag{39}$$

$$\text{RMSE} \sqrt{\sum_{q=1}^{5000} \left(\nabla \xi\right)^2 / 5000} \tag{40}$$

where $\xi_{\text{true}}$ is the true value of parameters; $\nabla \xi$ denotes the difference between the adjustment and true values.

For the situation where the a priori information is often disparate from the true accuracy or the a priori information is unknown in the actual measurement environment, the experiment is carried out in three cases. Case 1 is where nominal accuracy equals to true accuracy, while the second and third

are, respectively, a slightly differential spread of two pieces of information and an equal-weighted treatment in the case where the a priori one is unknown.

### 3.2. Real Data

Two experimental strategies are designed as follows:

1.    Self-calibration method based on nonlinear least-squares ignoring random errors;
2.    General self-calibration method.

We bring the nominal accuracy of the HDS3000 (4 mm for range and $3.3 \times 10^{-3\circ}$ for angles, one sigma) as a priori information into the adjustment process (Case 1), where the coordinates in the TS datum are treated as the true value, to analyze the validity and practicality of the algorithm proposed in this paper.

In addition, we have also assumed the case that the a priori information is unknown and the observations are treated with equal weights (Case 2). The self-calibration is implemented using the same two strategies described above, with the following results. It is assumed that the square root of the a priori variance component is $\sigma_0 = 1$.

In much of the literature, they directly take the nominal accuracy as a priori information to participate in the adjustment process, ignoring posterior estimation, which may lead to the observed values cannot match the appropriate weights and chop off the solution accuracy of the APs, nor can achieve perfect calibration results. Here, we use coordinate sequence in [28] as the experimental data, where eight points (five homonymous points using spherical target, and three checking points using planar target), tabulated in Table 2, are determined by the NET1200 (SOKKIA, 2003) and HDS3000 (Leica, 2003), respectively.

**Table 2.** Original 3D coordinates of 8 points.

| Classes | TLS (HDS3000) | | | TS (NET1200) | | |
|---------|---------|---------|---------|--------|---------|--------|
|         | $x$     | $y$     | $z$     | $X$    | $Y$     | $Z$    |
| Sphere1 | 3.8057  | −3.6132 | −0.4957 | 6.5368 | 10.0224 | 5.7071 |
| Sphere2 | 1.1437  | −6.5275 | −0.6502 | 2.7830 | 11.2521 | 5.5628 |
| Sphere3 | −0.6325 | −3.3331 | −0.6429 | 2.8041 | 7.5964  | 5.5640 |
| Sphere4 | −3.0580 | −3.7878 | −1.0332 | 0.4659 | 6.7989  | 5.1775 |
| Sphere5 | −3.4119 | −1.7673 | −0.6451 | 1.1509 | 4.8632  | 5.5611 |
| Plane1  | 1.6613  | −3.5856 | −0.5756 | 4.6813 | 8.9467  | 5.6292 |
| Plane2  | 0.7593  | −1.5648 | −0.5447 | 4.8888 | 6.7429  | 5.6555 |
| Plane3  | −1.7224 | −0.9954 | −0.5689 | 3.0013 | 5.0232  | 5.6314 |

## 4. Results and Discussions

### 4.1. Simulated Data

Since Strategy 1 (blue lines) does not consider the APs, so that they are not shown in (g)–(k) in Figure 4. From the results of Table 3 and Figures 4–9, we can find that,

1.    Strategy 1, without any error correction, has the largest deviation from the true value of the parameters, which leads to the necessity of instrument calibration, echoing [10,14–17,23,25];
2.    The results for both Strategy 2 (red lines) and Strategy 3 (green lines) hover around zero. However, compared to Strategy 3, the results of Strategy 2 are more scattered and diverged from the true value, even somewhat beyond Strategy 1 (Figures 4c, 6c and 8c). So, it is clear that Strategy 3 outperforms Strategy 2, especially for the translation parameters and axis errors.
3.    Strategy 3 allows for more accurate APs and EOPs. The RMSE of parameters in Strategy 3 is the closest to zero, taking into account the preservation of decimal places, and the accuracy of all

parameters is improved relative to Strategy 1 and 2. For the APs, except for the addition constant *m*, according to the order of the parameter vector (Equation (11)), Strategy 3 improves respectively by 2%, 48.1%, 30.9%, and 53.7%, with improving accuracy by 48.7% to 84.9% for EOPs of those in Strategy 2 in all three cases, demonstrating the validity and robustness of the general method.

4.  After the adjustment, the sequences of addition and multiplication constant in Strategy 2 and 3 are with broadly similar trends, probably due to the fact that the values of the two constants are much higher than the added random errors.

5.  In Strategy 3, the RMSE of the homonymous points are much smaller than those in Strategy 2, improving from $10^{-4}$ to $10^{-7}$, as shown from (a) to (f) in Figures 4, 6 and 8, but the remaining points are essentially the same and do not deviate greatly from each other. On the other hand, if no random errors are removed from the observations, the RMSE results of the homonymous points of Strategy 3 shared the same trend as Strategy 2.

**Table 3.** RMSE of parameter vector.

| S [a] | EOPs | | | | | | APs | | | | |
|---|---|---|---|---|---|---|---|---|---|---|---|
| | $\Delta x$ | $\Delta y$ | $\Delta z$ | $\varphi$ | $\omega$ | $\kappa$ | $m$ | $\lambda$ | $c$ | $i$ | $t$ |
| 1 | $3.9 \times 10^{-3}$ | $4.8 \times 10^{-3}$ | $7 \times 10^{-4}$ | $8 \times 10^{-4}$ | $4 \times 10^{-4}$ | 0.0115 | $-$ [c] | $-$ | - | - | - |
| 2 | $3 \times 10^{-4}$ | $3.5 \times 10^{-4}$ | $5.3 \times 10^{-4}$ | $1.2 \times 10^{-5}$ | $1.1 \times 10^{-5}$ | $3.5 \times 10^{-5}$ | $1.1 \times 10^{-3}$ | $5.7 \times 10^{-5}$ | $2.9 \times 10^{-5}$ | $1.8 \times 10^{-5}$ | $2.2 \times 10^{-5}$ |
| 3-C1 [b] | $4.8 \times 10^{-5}$ | $5.8 \times 10^{-5}$ | $1 \times 10^{-4}$ | | | | $1.8 \times 10^{-5}$ | $1.1 \times 10^{-3}$ | $5.6 \times 10^{-5}$ | $1.5 \times 10^{-5}$ | $1.3 \times 10^{-5}$ | $1.0 \times 10^{-5}$ |
| 3-C2 | $4.8 \times 10^{-5}$ | $5.8 \times 10^{-5}$ | $1 \times 10^{-4}$ | $6.1 \times 10^{-6}$ | $5.0 \times 10^{-6}$ | $1.8 \times 10^{-5}$ | $1.1 \times 10^{-3}$ | $5.6 \times 10^{-5}$ | $1.5 \times 10^{-5}$ | $1.3 \times 10^{-5}$ | $1.0 \times 10^{-5}$ |
| 3-C3 | $4.8 \times 10^{-5}$ | $5.8 \times 10^{-5}$ | $1 \times 10^{-4}$ | $6.1 \times 10^{-6}$ | $5.0 \times 10^{-6}$ | $1.8 \times 10^{-5}$ | $1.1 \times 10^{-3}$ | $5.6 \times 10^{-5}$ | $1.5 \times 10^{-5}$ | $1.3 \times 10^{-5}$ | $1.0 \times 10^{-5}$ |
| Ipv [d]/% | 84.9 | 83.5 | 79.8 | 48.7 | 56.5 | 49.6 | 0 | 2 | 48.1 | 30.9 | 53.7 |

[a] '**S**' represents the numerical order of strategies; [b] 'C' represents the numerical order of Cases; [c] '–' represents null value; [d] 'Ipv' denotes the improvements of Strategy 3 phase for Strategy 2 in percentage terms.

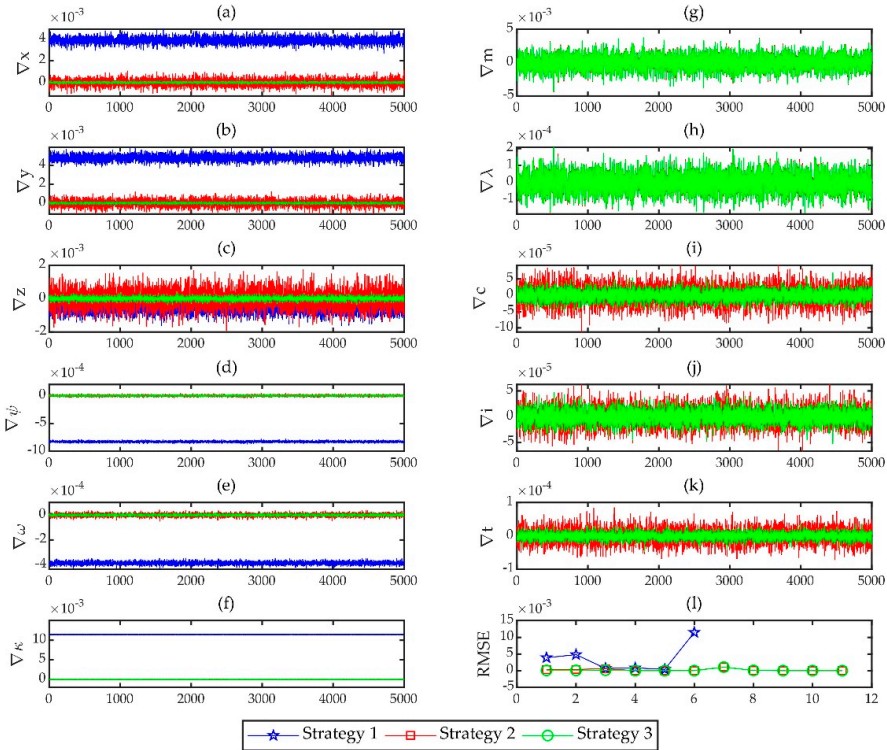

**Figure 4.** The difference in the sequences of the parameters in Case 1 with its RMSE where the horizontal axis indicates the experiment serial number from (**a**) to (**k**) and parameters' serial number in (**l**). (**a**)$\nabla x$, (**b**)$\nabla y$, (**c**)$\nabla z$, (**d**)$\nabla \varphi$,(**e**)$\nabla \omega$, (**f**)$\nabla k$, (**g**)$\nabla m$, (**h**)$\nabla \lambda$, (**i**)$\nabla c$, (**j**)$\nabla i$, (**k**)$\nabla t$, (**l**) RMSE of EOPs and Aps.

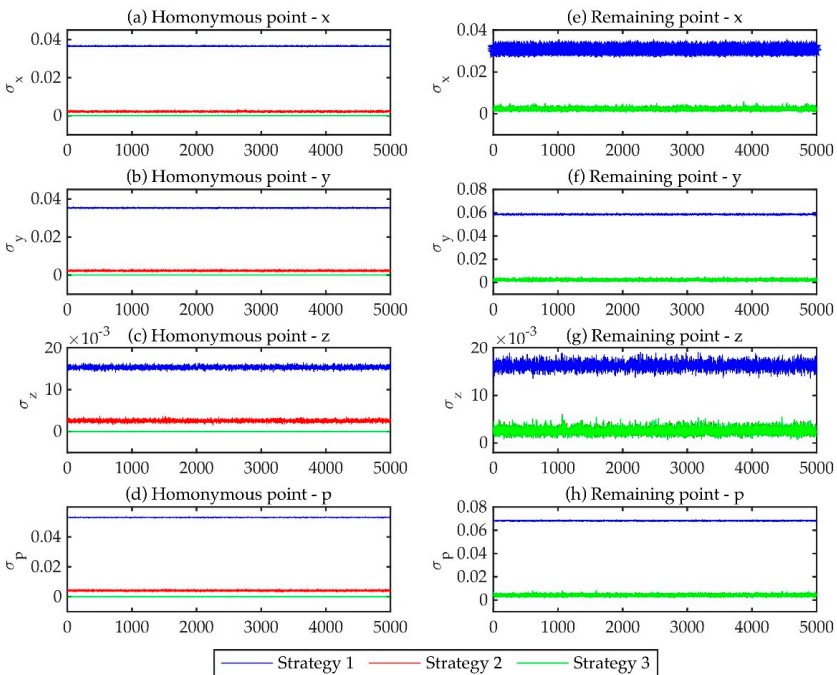

**Figure 5.** RMSE of points in Case 1, where the horizontal axis indicates the experimental serial number. (**a**) RMSE of homonymous points in $x$ direction; (**b**) RMSE of homonymous points in $y$ direction; (**c**) RMSE of homonymous points in $z$ direction; (**d**) positional RMSE of homonymous points; (**e**) RMSE of checking points in $x$ direction; (**f**) RMSE of checking points in $y$ direction; (**g**) RMSE of checking points in $z$ direction; (**h**) positional RMSE of checking points.

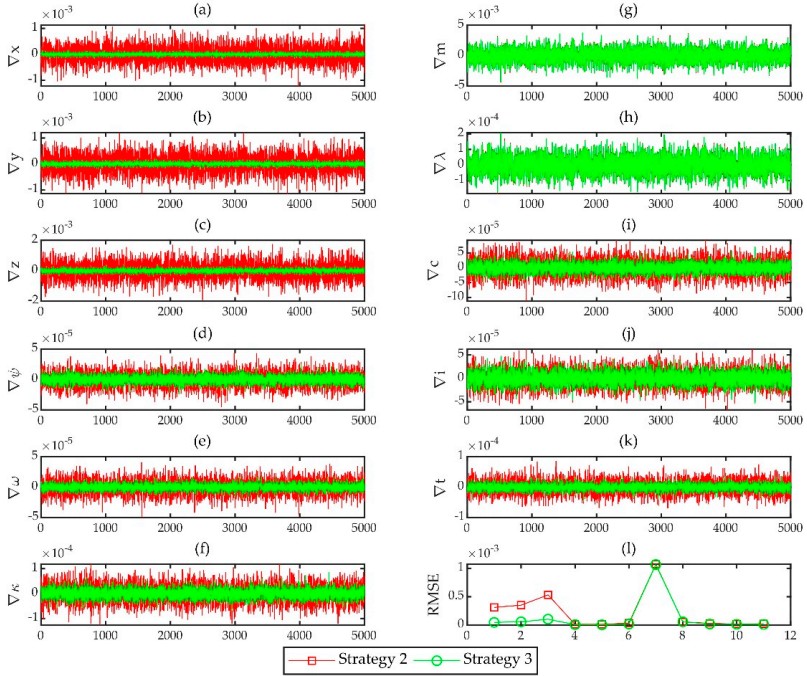

**Figure 6.** The difference sequences of parameters in Case 2 with its RMSE where the horizontal axis indicates the experiment serial number from (a) to (k) and parameters' serial number in (l). (**a**)$\nabla x$, (**b**)$\nabla y$, (**c**)$\nabla z$, (**d**)$\nabla \varphi$,(**e**)$\nabla \omega$, (**f**)$\nabla k$, (**g**)$\nabla m$, (**h**)$\nabla \lambda$, (**i**)$\nabla c$, (**j**)$\nabla i$, (**k**)$\nabla t$, (**l**) RMSE of EOPs and Aps.

It should be noted that the results for Strategy 1 are the same in all cases so that they are not repeated in Figures 6–9 for Case 2 and Case 3 for ease of reading. In addition, we also computed the correlation coefficient matrix of the unknown parameters after each loop acquisition and stipulated

the final correlation information by taking the average of 5000 experiments (removing the diagonal elements). We found that 55% of them decreased and 45% increased, and their magnitudes were in the order of $10^{-4}$ to $10^{-3}$, indicating that the general algorithm does improve the correlation of the parameters, but not very significantly, mainly due to the fact that the correlation of the parameters was ignored over the course of the data simulation.

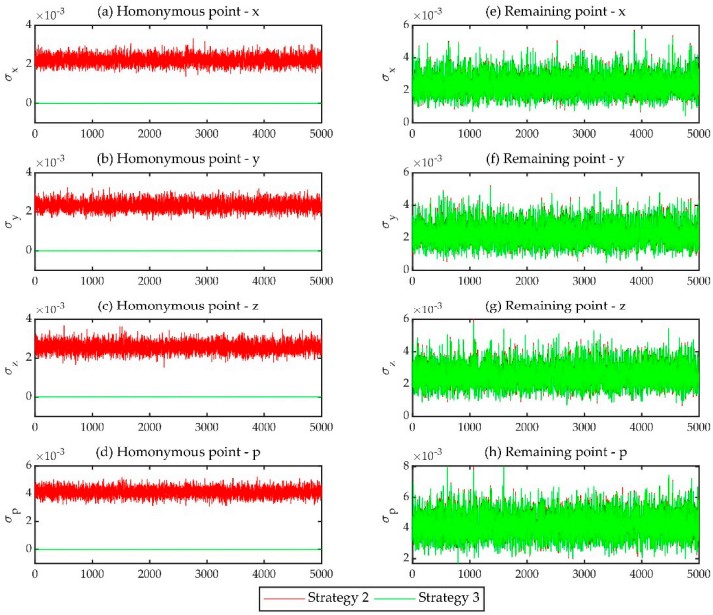

**Figure 7.** RMSE of points in Case 2 where the horizontal axis indicates the experimental serial number. (**a**) RMSE of homonymous points in $x$ direction; (**b**) RMSE of homonymous points in $y$ direction; (**c**) RMSE of homonymous points in $z$ direction; (**d**) positional RMSE of homonymous points; (**e**) RMSE of checking points in $x$ direction; (**f**) RMSE of checking points in $y$ direction; (**g**) RMSE of checking points in $z$ direction; (**h**) positional RMSE of checking points.

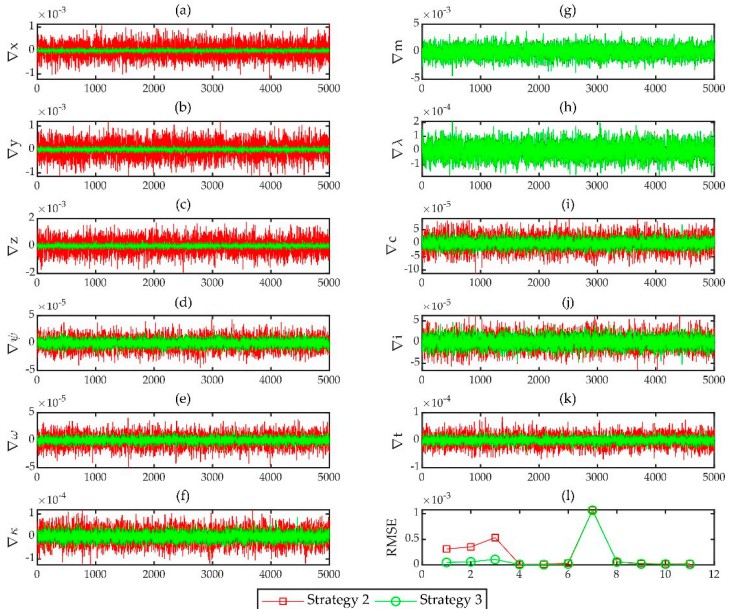

**Figure 8.** The difference sequences of parameters in Case 3 with its RMSE where the horizontal axis indicates the experiment serial number from (a) to (k) and parameters' serial number in (l). (**a**)$\nabla x$, (**b**)$\nabla y$, (**c**)$\nabla z$, (**d**)$\nabla \varphi$,(**e**)$\nabla \omega$, (**f**)$\nabla k$, (**g**)$\nabla m$, (**h**)$\nabla \lambda$, (**i**)$\nabla c$, (**j**)$\nabla i$, (**k**)$\nabla t$, (**l**) RMSE of EOPs and Aps.

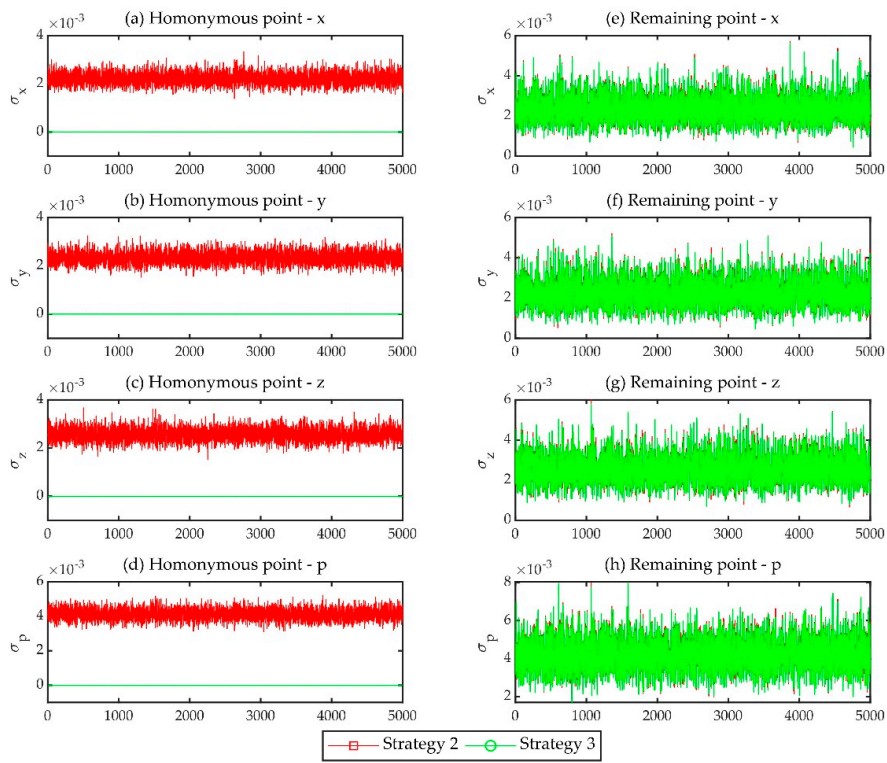

**Figure 9.** RMSE of points in Case 3 where the horizontal axis indicates the experimental serial number. (**a**) RMSE of homonymous points in $x$ direction; (**b**) RMSE of homonymous points in $y$ direction; (**c**) RMSE of homonymous points in $z$ direction; (**d**) positional RMSE of homonymous points; (**e**) RMSE of checking points in $x$ direction; (**f**) RMSE of checking points in $y$ direction; (**g**) RMSE of checking points in $z$ direction; (**h**) positional RMSE of checking points.

### 4.2. Real Data

Multiple scientific publications and the above simulation experiments have shown that a reasonable and correct instrument calibration can effectively weaken the effect of APs on the coordinate sequence and improve the accuracy of the coordinate data, see, e.g., [14,22,27,30]. Likewise, we validated the proposed method using real data and the results are shown in Table 4.

**Table 4.** Homonymous points accuracy in two Cases.

| Strategies | $\sigma_x$ | $\sigma_y$ | $\sigma_z$ | $\sigma_p$ |
|---|---|---|---|---|
| 1 | $1.62 \times 10^{-4}$ | $7.46 \times 10^{-5}$ | $5.39 \times 10^{-5}$ | $1.86 \times 10^{-4}$ |
| 2-C1 [1] | $6.88 \times 10^{-8}$ | $5.10 \times 10^{-8}$ | $1.42 \times 10^{-8}$ | $8.68 \times 10^{-8}$ |
| 2-C2 | $6.67 \times 10^{-7}$ | $5.33 \times 10^{-7}$ | $1.89 \times 10^{-9}$ | $8.54 \times 10^{-7}$ |

[1] 'C' represents the numerical order of cases.

From Table 4, we can see that the accuracy of the general self-calibration model is always the highest for the homonymous point part. For both cases, the level of point accuracy for the homonymous points could be increased from $10^{-4}$ to $10^{-8}$ and $10^{-7}$, respectively. Results in Table 4 also show that the proposed method can not only check the systematic errors, but also effectively remove the influence of random errors, and at the same time, it is also robust to different weighting methods. The results for the corrected checkpoints are similar to those of the simulation experiments due to the fact that the random error of the checkpoints cannot be estimated.

In fact, we still analyzed the correlations among the parameters, i.e., we calculated their correlation coefficient matrix based on the variance-covariance information of the unknown parameters from Strategy 1 and Strategy 2, respectively. As an example, Figure 8 shows the calculation of the absolute

difference between the correlation coefficients of strategy 1 and strategy 2 under the equal weighting circumstance, where the horizontal and vertical axes indicate the order of the parameters. The elements in Figure 10 greater than 0 denote that the correlation of the parameters in Strategy 2 was lower than in Strategy 1, and vice versa.

　　　We then performed a statistical analysis of the 110 elements of the difference matrix (without considering the diagonal elements), in which there were 74 elements greater than 0 and 36 elements less than 0, indicating that the general algorithm impairs most of the correlation among parameters; for those elements with increased correlations, it is conjectured that this may be due to the lack of use of geo-referencing.

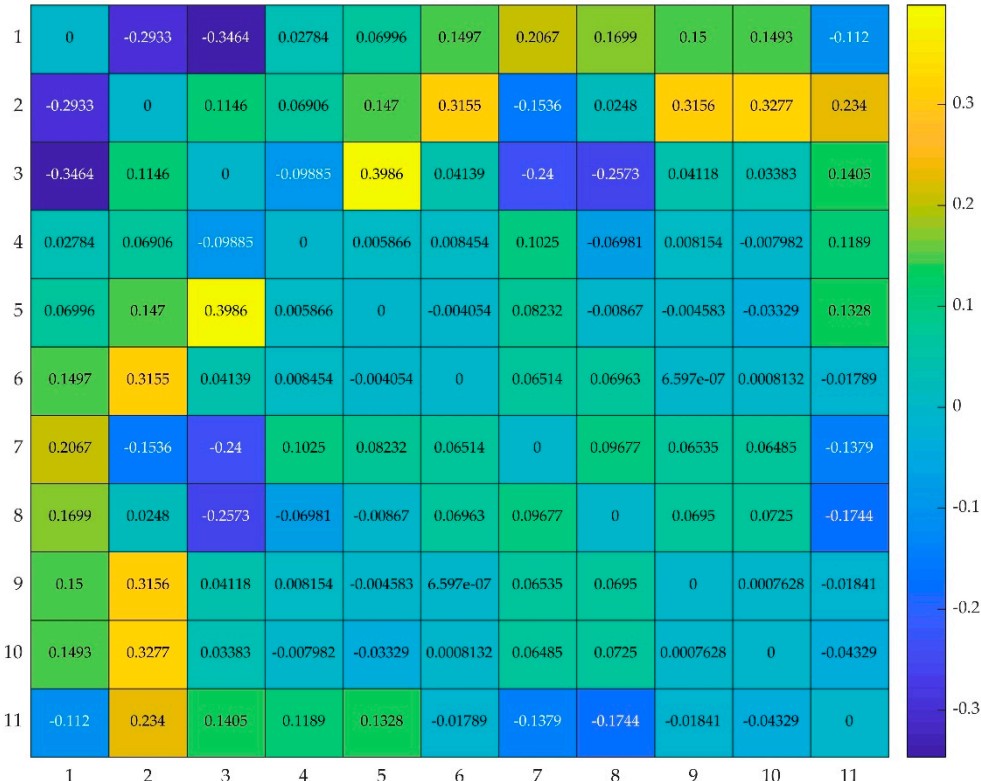

**Figure 10.** Differences in the correlation coefficients between Strategy 1 and Strategy 2.

## 5. Conclusions

　　　In this study, we proposed a general point-based self-calibration method for TLS taking into account both random errors in the observations and posterior estimates in the cases where the a priori information differed from the true accuracy. Therefore, it was theoretically more rational and rigorous than the traditional self-calibration methods. In cases where the nominal accuracy was different from the true accuracy, or where the a priori information was unknown, a posterior estimation could be performed to obtain a more realistic calibration parameter. The coordinate components and positional accuracy of the coordinate dataset after the general method processing was be effectively improved, and more importantly, the difference between the corrected coordinates and the true coordinates was closer to a straight line, indicating that the general method wasmore stable and robust, as shown in the left four panels of Figures 4, 6 and 8. The coordinate difference between the true and corrected values obtained by the general method was sometimes larger compared to the traditional method because the random error of the coordinates of the remaining points (checkpoints) could not be effectively estimated, but this did not determine the results of the calculation of EOPs and APs, or the validity of the general calibration method.

　　　In this paper, the raw observations were divided into two categories, distance and angle, in the posterior estimation process. Due to the environmental or human interference, the same kind of

observations may have different a priori information [40,41], that is, observation values needed to be divided into more categories, and the idea in this paper can be extended to address this problem. The algorithm in this paper is derived based on the observation equation of TLS and the GH model, thus it could also be applied to solve some other problems, such as point cloud registration, coordinate transformation, image processing, etc.

The correlation among most of the parameters was weakened by attaching weights to the observations, about 67%, but the correlation was undeniably increased in some other locations (see Figure 10). Also, it has been proposed in some literature [19] that the rational introduction of the geo-referencing and network design could effectively weaken the parameter correlation, which is the next step of research.

**Author Contributions:** All authors contributed to conceptualization, methodology, computation, investigation, visualization, writing-original draft preparation, reviewing and editing aspects of the study. All authors have read and agreed to the published version of the manuscript.

**Funding:** This research was funded by National Natural Science Foundation of China, grant number 41974213, and the APC was funded by 41974213.

**Acknowledgments:** We thank the anonymous reviewers for their comments that improved the manuscript.

**Conflicts of Interest:** The authors declare no conflict of interest.

## Appendix A

It should be reminded that since the SOKKIA NET 1200 and the Leica HDS3000 have different reference coordinate systems, where the first one is a right-handed system and the second one is a left-handed system. Thus, the rotation matrix needs to be changed accordingly during the adjustment process.

The basic idea of the VCE algorithm is to weight the observations according to the a priori information, and then calculate the squared sum of the residuals of the observations, and finally estimate the variance of the observations according to a certain principle. For this reason, the order of the elements needs to be rearranged to classify the observations.

Depending on the conditions applicable to VCE, the weight matrix $P$ can be written as:

$$P = blkdiag\left( \frac{\sigma_0^2}{\sigma_s^2} E_{\text{n}\times\text{n}}, \frac{\sigma_0^2}{\sigma_\theta^2} E_{\text{n}\times\text{n}}, \frac{\sigma_0^2}{\sigma_\alpha^2} E_{\text{n}\times\text{n}} \right) \tag{A1}$$

Subjecting to the observation equation of TLS, i.e., Equation (13), the partial differential for $q$th observation values can now be obtained from:

$$\begin{cases} dx_q = a_q^1 ds_q + a_q^2 d\theta_q + a_q^3 d\alpha_q \\ dy_q = a_q^4 ds_q + a_q^5 d\theta_q + a_q^6 d\alpha_q \\ dz_q = a_q^7 ds_q + a_q^8 d\theta_q + a_q^9 d\alpha_q \end{cases} \tag{A2}$$

where $a_q^{1,2\cdots9}$ refers to the coefficient for each partial differential, $q$ is the sequence number of observations. This can be written in a matrix notation as:

$$\begin{bmatrix} dx_1 \\ dy_1 \\ dz_1 \\ \vdots \\ dx_j \\ dy_j \\ dz_j \end{bmatrix} = \begin{bmatrix} a_1^1 & a_1^2 & a_1^3 \\ a_1^4 & a_1^5 & a_1^6 & & 0 \\ a_1^7 & a_1^8 & a_1^9 \\ & & \ddots \\ & & & a_j^1 & a_j^2 & a_j^3 \\ & 0 & & a_j^4 & a_j^5 & a_j^6 \\ & & & a_j^7 & a_j^8 & a_j^9 \end{bmatrix} \begin{bmatrix} ds_1 \\ d\theta_1 \\ d\alpha_1 \\ \vdots \\ ds_j \\ d\theta_j \\ d\alpha_j \end{bmatrix} \tag{A3}$$

Due to the changes in the order of the weight matrix and observation vectors, the above consistent equation system could be converted into a new form

$$
\begin{bmatrix}
dx_1 \\
dy_1 \\
dz_1 \\
\vdots \\
dx_j \\
dy_j \\
dz_j
\end{bmatrix}
=
\underbrace{
\begin{bmatrix}
a_1^1 & 0 & \cdots & a_1^2 & 0 & \cdots & a_1^3 & 0 & \cdots \\
a_1^4 & 0 & \cdots & a_1^5 & 0 & \cdots & a_1^6 & 0 & \cdots \\
a_1^7 & 0 & \cdots & a_1^8 & 0 & \cdots & a_1^9 & 0 & \cdots \\
 & \ddots & & & \ddots & & & \ddots & \\
0 & \cdots & a_j^1 & 0 & \cdots & a_j^2 & 0 & \cdots & a_j^3 \\
0 & \cdots & a_j^4 & 0 & \cdots & a_j^5 & 0 & \cdots & a_j^6 \\
0 & \cdots & a_j^7 & 0 & \cdots & a_j^8 & 0 & \cdots & a_j^9
\end{bmatrix}}_{B}
\underbrace{
\begin{bmatrix}
ds_1 \\
\vdots \\
ds_j \\
d\theta_1 \\
\vdots \\
d\theta_j \\
d\alpha_1 \\
\vdots \\
d\alpha_j
\end{bmatrix}}_{de}
\tag{A4}
$$

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
