# Peer review of "A General Point-Based Method for Self-Calibration of Terrestrial Laser Scanners Considering Stochastic Information"

_remotesensing, doi:10.3390/rs12182923_

Round 1
Reviewer 1 Report
In this work the authors propose a method for self-calibration of terrestrial laser scanner considering stochastic Information. This manuscript will be interested to the readers Remote Sensing. However, the manuscript needs minor revision before it can be accepted for publication.
General comments:
- A flowchart describing the whole procedure proposed by the authors will be useful for the readers to have a quick overview of this work.
- Equations needs to be revised for mathematical consistency and notation.
- Use a summary table with the parameters used in the tests (lines 298-299)
- The figures of the test case of the simulated data are missing.
- The lines 311-328 should be included in the methods section.
- Discussions section is missing. In the Discussions section “strong points” and the "limits" of the study should be mentioned.
- In the Conclusions section don’t use the word conclusion, avoid writing the conclusions in points.
Reviewer 2 Report
The global aim of this paper is to propose a self-calibration method for Terrestrial Laser Scanners. This, in view of the results given, improves the coordinate measurements by taking random errors into consideration.
Personally speaking, I would say that the reduction of the point uncertainty after calibration would improve the way the object surface coordinates are represented in space, as well as improve point cloud segmentation. All this has a clear impact on the quality of the object scanned for subsequent as-built 3D modelling.
On the other hand, there are certain aspects to be addressed so that the paper could be published.
The following comments may be useful to improve the paper:
- From my point of view, the paper's structure should be revised.
- The abstract should be improved.
- Please include additional references on the observation principle of TLS.
- Please cite any software or equipment mentioned, such as Callidus 1.1, Leica HDS 3000 and Leica HDS 2500, FARO 880, MatlabR2019b, etc. This is applicable to the whole manuscript.
- Please exhaustively revise the English grammar and punctuation, as well as the correct use of complementary comments for the reader, such as "for more information, refer to...", "see [a,b,c]", etc.
>>>> TITLE
Taking into account the paper, the title should be "A General Point-based Method for Self-Calibration of Terrestrial Laser Scanners Considering Stochastic Information". (Scanners, in plural).
>>>> ABSTRACT
The structure of the abstract should be enhanced by firstly describing the context and research problem. Please note that 80% of the abstract is methodology, and 20% is related to conclusions.
The novelty of this research should also be highlighted.
The conclusions could be reiforced at the end of this abstract.
Quantitative data should be included at the end of the abstract to highlight the importance of this research against other research in the field, as well as to justify the Authors' statement "the proposed method can properly estimate the additional parameters with high precision".
The limitations of this article should be described.
>>>> INTRODUCTION
Please rewrite line 52-53; it is not clear: "As for the first approach, it is by..."
Please define 'AM-CW' for the readership of this journal.
Please cite any software or equipment mentioned, such as Callidus 1.1, Leica HDS 3000 and Leica HDS 2500, FARO 880, MatlabR2019b, etc. This is applicable to the whole manuscript.
The novelty of this paper against other studies on the topic should be clearly described in one or two sentences.
Please include a final paragraph in this Introduction section to explain how the paper is organised.
>>>> 2. SELF-CALIBRATION MODEL
References should be included to support the observation principle of TLS, included in this paper.
Please rewrite lines 118-121 for the sake of clarity; please check grammar here, as 'is' should be 'are' in line 120.
>>>> 3. DERIVATION OF GENERAL SELF-CALIBRATION MODEL
Lines 180-181: "Due to the nonlinear nature of the Equation (8). The Gauss-Newton method [41] of non-linear LS is adopted to derive the solution". Should not the first phrase and the second sentence constitute a full sentence? Please rewrite this.
>>>> 4. EXPERIMENTS AND RESULTS
The experimental approach, lines 282-333 for 'Simulated data' and lines 401-416 for 'Real data', from my point of view, should constitute a new section 4. 'Experimentation' or similar, since the conditions of the tests are especifically described and defined.
Next, accordingly, the figures, tables, and data interpretation out of those lines 282-333 and 401-416 should constitute section 5. 'Results and Discussion'. Thereby, the paper's structure may turn more gradual, i.e., first, we define the conditions/methods, next, the results are shown and discussed.
>>>> CONCLUSIONS
The conclusions are well structured and complete, since they include the methodological approach, results, implications and research limitations.
>>>> WRITING AND ENGLISH LANGUAGE
Please revise this section carefully throughout the manuscript, not only on what is specifically mentioned here.
Please exhaustively revise English grammar and punctuation.
Line 54: "Gielsdorf et al. pioneered the concept of TLS calibration for low-cost scanner". It should be "for low-cost scanners", in plural.
Lines 62-63: "to weaken the correlation among the parameters, referring to [20-21] to find more details". Those two clauses cannot be put together using a comma. Please avoid comma splice. Use semicolon or full stop instead. Please note that the first clause is intended to describe the purpose of the sentence, but the second clause is directed to the reader of this paper.
Line 72: "A/THE point-based self-calibration method often REQUIRES..."; both article and plural form needed.
Line 73: "Alternatively, self-calibration based on planes, that have been proposed by [12, 25-29] <<<<aroused as wish>>>>". Please rewrite this sentence to avoid the last expression.
Lines 75-79: Please make sure the message reaches the reader here. A five-line sentence may not be effective in that sense.
Line 79: "a few of literature". Line 401: "Numerous literature". 'Literature' in uncountable; therefore, cannot be put together with 'a few' or 'numerous'. Instead, the Authors could use 'certain' or 'multiple' scientific publications.
Lines 81-82: "As a result, for the existing calibration models that are not theoretically rigorous or inadvertently increase the complexity of the solution". There is no subject in this sentence. Please rewrite it. It is not clear enough.
Espressions such as "referring to [39, 40] et al., for more information" or similar should not be part of the sentences they follow, but put between brackets. Please exhaustively revise this throughout the manuscript, as the content of those expressions and that of the preceding scientific sentence are not the same.
Line 182: "appropriate approximate values (initial values) of e is". It should be "appropriate approximate values (initial values) of e ARE", in plural. Line 120 'is', line 121 'the scanner IS based' or 'the scannerS are based'. Line 190: 'is' should be 'are'. Please carefully revise the accordance of the subject with the number in the verb throughout the manuscript.
Line 277: "it's" should not be contracted.
Line 415: "are treat" should be "are treated".
Lines 451-452: 'more' is redundant (used twice). Please check it.
Round 2
Reviewer 2 Report
All my concerns have been addressed by the Authors. There are just a few more minor changes to make so that the paper could be published (reinforcement of implications and minor language issues).
Note 1: this review is based on the 'Track changes' version of the manuscript (attached), so the line numbers may change from the 1st round version of the paper submitted by the Authors.
Note 2: the line numbers in this version start with no.43.
>>>> ABSTRACT
Line 56: "Due to the existence of instrument itself, environment or human factors, ..." could be "Due to the existence of environment<<<AL>>> (ENVIRONMENTAL) or human factors, and because of the instrument itself, ..." to ease comprehension.
Line 66: "Considering THAT the proposed method is a non-linear..."
>>>> IMPLICATIONS (no especific section)
Line 597-598: for the sentence "..., and to provide a basis for improving subsequent point cloud segmentation, 3D modelling" the Authors should include ", etc." at the end of it.
In fact, the implication stated in this paper about an improvement in 3D modelling, could be reinforced by citing further research publications on the accuracy evaluation/assessment of point cloud data and 3D meshes for 3D reconstruction (already in Remote Sensing citation style):
- Antón, D.; Pineda, P.; Medjdoub, B.; Iranzo, A. As-Built 3D Heritage City Modelling to Support Numerical Structural Analysis: Application to the Assessment of an Archaeological Remain. Remote Sens. 2019, 11, 1276. https://doi.org/10.3390/rs11111276
-Jindian Liu, Qilin Zhang, Jie Wu, Yuchao Zhao,Dimensional accuracy and structural performance assessment of spatial structure components using 3D laser scanning,Automation in Construction,2018, 96, P: 324-336.
ISSN 0926-5805,https://doi.org/10.1016/j.autcon.2018.09.026.
(http://www.sciencedirect.com/science/article/pii/S0926580518301699)
I strongly believe that this could reinforce the implications of this paper. The accuracy of point cloud data is decisive for meshing quality in diverse fields (heritage, building and constructions, arts, etc.).
>>>> WRITING AND ENGLISH LANGUAGE
Espressions such as "referring to [20, 21] to find more details" or similar should not be part of the sentences they follow, but put between brackets. The content of those expressions and that of the preceding scientific sentence are not the same but "superfluous" as stated by the Authors in the conver letter. I completely agree; they should be removed.
Line 240: "In addition, there are still multiple literature". 'Literature' should be changed for 'scientific publications' here, as the former is uncountable. The same happens in line 1076 (4.2. Real data).
Please unify: "modelling" or "modeling". The language region should be consistent.
